# Dietary *Schizochytrium* sp. Meal Enhances the Fatty Acid Profile in Pirarucu (*Arapaima gigas*) Fillets with No Effect on Growth Performance and Health Status

**DOI:** 10.3390/ani15050712

**Published:** 2025-03-02

**Authors:** André D. Nobre, Renata V. Mendonça, Ana Beatriz de S. Farias, Fernando Y. Yamamoto, Ligia U. Gonçalves

**Affiliations:** 1Graduate Program in Animal Science and Fisheries, Federal University of Amazonas, Manaus 69067-055, AM, Brazil; nobreandre00@gmail.com; 2Veterinary Medicine, Federal Institute of Amazonas, Manaus 69083-475, AM, Brazil; verasr990@gmail.com; 3Graduate Program in Aquaculture, Nilton Lins University, Manaus 69058-030, AM, Brazil; anabeatrizz.sena@gmail.com; 4Thad Cochran National Warmwater Aquaculture Center, Mississippi Agriculture and Forestry, Experiment Station, Mississippi State University, Stoneville, MS 38776, USA; 5Department of Wildlife, Fisheries and Aquaculture, College of Forestry Resources, Mississippi State University, MS 39762, USA; 6National Institute for Amazonian Research, Manaus 69067-375, AM, Brazil

**Keywords:** *Arapaima gigas*, docosahexaenoic acid, n-3 fatty acid, *Schizochytrium* sp. meal

## Abstract

This study focused on enhancing omega-3 in fillets of pirarucu by supplementing *Schizochytrium* sp. meal as a source of docosahexaenoic acid. Two diets with soybean oil or 5% *Schizochytrium* sp meal were tested for pirarucu. The feeds were readily accepted and no mortality was observed. The fish did not present significant differences in growth performance or hemato-biochemical parameters, except for hemoglobin and total protein. The use of 5% *Schizochytrium* sp. meal is a viable ingredient for enhancing omega-3, mainly docosahexaenoic acid, in fillets of pirarucu. However, further studies are recommended to periodically analyze the fillet fatty acid profile to determine the optimum time point for omega-3 absorption.

## 1. Introduction

Fishmeal and fish oil are the primary sources of n-3 highly unsaturated fatty acids (n-3 HUFAs), such as eicosapentaenoic acid (EPA) and docosahexaenoic acid (DHA). In aquafeeds, their use is being increasingly viewed as unsustainable due to the overfishing of wild forage marine fish and the competition between food for human consumption and for aquaculture feed manufacturing [1]. In order to mitigate this issue, plant-based ingredients have been utilized as replacements for fish oil in aquafeeds to reduce reliance on marine resources [2]. While these alternatives provide the essential energy and protein required for fish growth, they generally lack the n-3 HUFA, leading to lower EPA and DHA levels in fish fillets and reducing their nutritional benefits to consumers [3]. Eicosapentaenoic acid (20:5n-3; EPA) contributes to reducing inflammatory processes, insulin, and biosynthesis of triglycerides, as well as improving cardiovascular health [4]. Docosahexaenoic acid (22:6n-3; DHA), however, is essential to the formation and functioning of the central nervous system and retina, and prevents degenerative diseases (e.g., Alzheimer’s) [5,6]. The reduction in EPA and DHA in the fillets of fish has stimulated the urgent need for more sustainable and nutritionally viable alternatives to fish oil that can support both the productivity of aquaculture and the nutritional quality of fish fillets.

To address this concern, *Schizochytrium* sp. has been successfully used as a source of DHA in aquafeeds. *Schizochytrium* sp. is a heterotrophic unicellular protist from the *Thraustochytriaceae* family [7] capable of producing high levels of lipids (~55%) [8], and DHA is one of the primary fatty acids (up to 60% of total lipids) [9,10,11]. Unlike photosynthetic microalgae, *Schizochytrium* sp. can be efficiently cultured in industrial bioreactors without requiring light or carbon dioxide, reducing production costs and environmental impact [12]. Its cultivation requires fewer resources, such as land and water, and can be performed in a controlled environment, reducing its impact on ecosystems [13]. The biological effects of *Schizochytrium* sp. are dose- and species-specific [14]; therefore, optimal levels must be studied for different species of interest for aquaculture. The use of *Schizochytrium* sp. meal has been thoroughly investigated, and successful complete fish meal and fish oil replacements have been reported for different fish species, such as Atlantic salmon (*Salmo salar*) [15], Nile tilapia (*Oreochromis niloticus*) [16], tambaqui (*Colossoma macropomum*) [17], and pink cusk-eel (*Genypterus blacodes*) [14].

Pirarucu (*Arapaima gigas*) is one of the Neotropical species of interest in South American aquaculture due to its phenotypical characteristics, such as rapid growth. Farmed juveniles weighing between 0.5 and 1.0 kg can reach 8 to 10 kg within a single production cycle of one year [18]. In addition, this fish has a ~57% fillet yield, absence of intramuscular bones in the fillet, and a mild flavored meat [19]. Although the DHA content is similar between wild (10.63 mg g^−1^) [20] and farmed pirarucu (11.2 mg g^−1^) [21], the absolute amount of DHA in wild fish fillets is greater due to their higher lipid content, approximately 8.26%, which is three times higher than that found in farmed fish fillets.

Currently, there is no specific commercial diet for pirarucu, and farmers rely on generic carnivorous fish feeds, which are typically rich in fish meal and fish oil. To enhance sustainability, the aquaculture industry has been shifting to plant-based alternatives, but this approach raises environmental concerns, including deforestation and high water consumption for irrigation. Moreover, pirarucu has limited ability to digest plant-based proteins, such as soybean meal [22], further complicating the transition to plant-based feeds.

Additionally, studies with Atlantic salmon [2] indicate that plant-based diets can lower omega-3 (EPA + DHA) levels in fish fillets, reducing their nutritional benefits. In this context, *Schizochytrium* sp. meal emerges as a superior DHA source, as it can be sustainably produced in industrial fermenters, minimizing pressure on fish stocks and land use. Fillet DHA concentrations can be increased by ~290% in *Colossoma macropomun* compared to soybean oil as the primary lipid source, reinforcing its potential to enhance the sustainability and nutritional value of aquaculture diets [17].

The fatty acid composition of the fillet reflects that of the diet; hence, including DHA-rich ingredients, such as *Schizochytrium* sp. meal, into the feed formulation for pirarucu may be an effective way to enhance the DHA content in their fillets. Therefore, this study aimed to evaluate the inclusion of *Schizochytrium* sp. in the diet of juvenile pirarucu as a replacement for soybean oil and its effects on the growth performance, health status, and fatty acid composition of the fillet.

## 2. Materials and Methods

This study was approved by the Ethics Committee on Animal Experimentation and Research of the National Institute for Amazonian Research (INPA), Manaus, Amazonas, Brazil under Protocol No. 230/2022.

### 2.1. Experimental Diets

Two diets were formulated to be isonitrogenous (40% crude protein), isolipidic (9%), and isoenergetic (17.65 MJ kg^−1^), one containing 5% *Schizochytrium* (5 SZ) and the other serving as a control with 0% *Schizochytrium* (0 SZ) (Table 1). All the ingredients were analyzed for proximate composition prior to formulating the experimental diets. The ingredients were ground, homogenized, hydrated with 27% of the volume of total feed weight, and posteriorly extruded in 6 mm pellets using a single screw extruder (INBRAMAQ, MX-80, São Paulo, SP, Brazil). The experimental pellets were immediately dried using a forced air oven at 55 °C for 24 h and immediately stored in black polyethylene bags in a freezer at −20 °C to prevent nutrient degradation and photooxidation.

### 2.2. Feeding Trial

Two weeks before the commencement of the feeding trial, the fish were stocked and fed the control diets in order to acclimate them to the experimental diet and the rearing conditions. The feeding trial was performed at the INPA Aquaculture Experimental Station, Manaus, Amazonas, Brazil. The trial followed a completely randomized design with two treatments (0 SZ and 5 SZ) and five experimental units (tank) per treatment. One hundred fish were individually weighed and measured (561.3 ± 13.4 g; 47.50 ± 7.07 cm), tagged using microchips in the dorsal muscle (AnimallTAG^®^—Korth RFID Ltd.a, São Carlos, SP, Brazil), and equally distributed in each experimental unit (10 fish/tank). The fish were stocked in 10 fiberglass tanks (1000 L) operating as a recirculating aquaculture system with phytoremediation for biological filtration. The fish were fed to satiety three times a day (8 a.m., 12 p.m., and 4 p.m.) for 12 weeks.

At the end of the feeding trial, all the fish from each tank were weighed and measured to compute the growth performance parameters as follows:Survival (S %) = (final number of fish × 100)/initial number of fish;Feed intake (FI, g) = feed offered/number of fish;Weight gain (WG, g) = final weight – initial weight;Feed conversion rate (FCR) = feed offered/weight gain;Relative growth rate (RGR, %/day) = (eg^−1^) × 100; e = Euler’s Number, g = [(ln final weight – ln initial weight)/Δt];Protein efficiency ratio (PER) = (weight gain (g)/consumed crude protein (g).

### 2.3. Sampling Process

The facility had a natural photoperiod (12:12 h light/dark), and the water quality parameters were measured during the acclimatization and feeding trial period, as described below: dissolved oxygen (5.15 ± 0.40 mg L^−1^), temperature (28.2 ± 0.6 °C), and pH (6.05 ± 0.50) were measured once a day at 09:30 am using a digital multiparameter probe (YSI, ProODO, Yellow Sprins, OH, USA). Total ammonia-nitrogen (0.33 ± 0.16 mg L^−1^) and total nitrite-nitrogen (0.50 ± 0.2 mg L^−1^) were measured once a week using colorimetric (Alphakit AT 101, Florianópolis, SC, Brazil). The water quality data throughout the feeding trial was suitable for what has been established for this species [24,25].

On the last day of the feeding trial, after weighing, seven fish from each tank were anesthetized by immersion in a eugenol solution (100 mg L^−1^) and euthanized by spinal cord rupture. Four of these fish were subjected to a procedure that included bleeding through the caudal vasculature (using 3 mL heparinized syringes), fillet sampling from each fish was labeled and immediately frozen using liquid nitrogen and then placed in a −80 °C freezer. The total lipids of the *Schizochytrium* sp. meal, diets, and fillets were determined using the Bligh and Dryer method [26]. The desiccation of intraperitoneal fat, liver, and total viscera was performed to determine conditional indices; the corresponding formulas for these indices are presented below. The remaining three fish were immediately euthanized (using the procedure described above) to determine the whole-body proximate composition.
Hepatosomatic index (HSI) (%) = [(liver weight (g)/body weight (g))] × 100;Intraperitoneal fat index (IPF) (%) = [(fat weight (g)/body weight (g))] × 100;Viscerosomatic index (VSI) (%) = [(viscera weight (g)/body weight (g))] × 100.

### 2.4. Hematological Analyses

The hemoglobin concentration (Hb) was determined using the cyanmethemoglobin method using a commercial kit (Labtest^®^, Vista Alegre, Lagoa Santa, MG, Brazil). The hematocrit (Ht %) percentage was evaluated using the microhematocrit technique. Erythrocyte (RBC × 10^6^ cells. µL^−1^) counts were performed using a hemocytometer (10 μL of blood, 2.0 mL of citrate formaldehyde).

The quantification of glucose was performed using a blood glucose meter (G-TECH^®^, Accumed Produtos Médico Hospitalares Ltd.; Duque de Caxias, RJ, Brazil) immediately after the collection of blood samples [27]. The whole blood was centrifuged (3000× *g* for 10 min at 4 °C) to obtain the plasma. These plasma samples were utilized for the assessment of cholesterol, high-density lipoprotein (HDL), and triglyceride levels via colorimetric enzymatic technique; total plasma proteins through the biuret reaction; albumin using the bromocresol green reaction; and globulin using the equation (total protein – albumin). These analytical procedures were conducted employing commercially available kits (InVitro Diagnóstica, Belo Horizonte, MG, Brazil) in conjunction with a spectrophotometer (HACH, DR6000, Loveland, CO, USA).

### 2.5. Fillet Fatty Acid Analyses

The *Schizochytrium* sp. meal, diets, and fillet samples were subjected to a fatty acid methyl ester (FAME) analysis following the methodology described by Santos-Júnior et al., 2014 [28]. The separation of methyl esters was performed via gas chromatography employing a gas chromatograph (Trace Ultra, Thermo Scientific, Waltham, MA, USA) equipped with a flame ionization detector and a fused-silica capillary column (100 m × 0.25 mm id, 0.25 µm cyanopropyl, CP-7420 Select Fame). The operational parameters were set as follows: detector temperature at 240 °C, injection port at 230 °C, column starting at 165 °C for 18 min, subsequently increasing by 4 °C/min to 235 °C, held for 14.5 min. Hydrogen was used as the carrier gas at 1.2 mL/min, nitrogen as the make-up gas at 30 mL/min, and a 1:80 split injection ratio was applied. For the purpose of identification, the retention times of the fatty acids were juxtaposed with those of standard methyl esters (Sigma, St. Louis, MO, USA). The software Chronquest 5.0 was used to automatically calculate the retention times and peak area percentage. The quantification of fatty acids (% of total lipids) was carried out utilizing tricosanoic acid (23:0) methyl ester (Sigma-Aldrich, Rockville, MD, USA) as an internal standard [29].

### 2.6. Statistical Analysis

The growth performance parameters, hematological analyses, condition indices, and fillet fatty acid data were subjected to the Shapiro–Wilk and Levene tests to validate normality and homoscedasticity, respectively. The hepatosomatic index data were rank-transformed to meet parametric assumptions. The data were further analyzed using an independent sample *t*-test, with Student’s *t*-test being employed to compare the two groups (0 SZ and 5 SZ) (*p* < 0.05) using Statistica 13.3 (TIBCO Software Inc., Santa Clara, CA, USA).

## 3. Results

### 3.1. Growth Performance

Although dominance interactions were observed during the experimental period, the experimental diets were readily accepted by the pirarucu. During the entire experimental period, only one dead fish was recorded. There was no significant difference in growth performance and the somatic indices (*p* > 0.05) (Table 2). The fish grew by around 291.2%, with an average feed conversion ratio of 1.58.

### 3.2. Hematology Analyses

The hematocrit, glucose, albumin, globulin, cholesterol, cholesterol HDL, and triglycerides were not significantly different between the experimental groups during the 12-week feeding period (Table 3). However, the hemoglobin of the fish fed the 5 SZ diet was higher than those fed the 0 SZ diet, and the total protein was lower for the fish fed with 5 SZ (*p* < 0.05).

### 3.3. Fillet Fatty Acid Composition

As shown in Table 4, no significant differences were observed between the experimental groups. Saturated fatty acids (SFAs) and monounsaturated fatty acids (MUFAs) were predominant in all the pirarucu fillets analyzed. Palmitic acid (16:0), stearic acid (18:0), and oleic acid (18:1 n-9) showed the highest concentrations of SFA and MUFA, respectively, for both experimental groups. Palmitic acid presented a 20% greater concentration in the fillets of the pirarucu fed the 5 SZ diet when compared to those fed the 0 SZ diet. On the other hand, those fed the 5 SZ diet presented a reduction in the stearic and oleic concentrations of 9.89% and 20.82%, respectively.

The 5 SZ feed reduced α-linolenic acid—ALA (18:3 n-3) and linoleic acid—LA (18:2 n-6) concentrations by 32.60% and 29.33%, respectively. The concentration of eicosapentaenoic acid—EPA (20:5 n-3) and docosahexaenoic acid—DHA (22:6 n-3) in the 5 SZ diet impacted their retention in the fillets of the pirarucu. The EPA and DHA concentrations in the fillets of the pirarucu were 59.82% and 845.54% higher in the fish fed the 5 SZ diet when compared to those fed 0 SZ. In addition, the use of 5% of *Schizochytrium* sp. meal in the diets of the juvenile pirarucu increased the n-3/n-6 ratio by 457.69%.

## 4. Discussion

The fish fed the two experimental diets presented similar growth performance, which can be attributed to the isoproteic, isolipidic, and isocaloric composition of the diets. This finding aligns with previous studies reporting no adverse effects on growth performance when the fish fed the diets with the inclusion of 5% *Schizochytrium* sp. meal, as observed in Atlantic salmon (*Salmo salar*) [15], pink cusk-eel (*Genypterus blacodes*) [14], and tambaqui (*Colossoma macropomum*) [17]. Moreover, although no specific essential fatty acid requirements have been established for pirarucu, the experimental diets provided more than 0.5–1.0% of LA, LNA, and ARA, as recommended by the National Research Council [23] for freshwater carnivorous fish. The fish are unable to synthesize linoleic acid and α-linolenic acid de novo, which is why these are known as essential fatty acids and must be provided in the diet to prevent nutritional deficiencies [30,31]. Marine fish require EPA and DHA in feeds, as they are unable or have a limited capacity to biosynthesize these fatty acids de novo from short-chain precursors [23,30,31]. Unlike marine fish, freshwater fish can convert linoleic acid into arachidonic acid and linolenic acid into EPA, and eventually to DHA, through a sequential process of desaturation and elongation involving three key enzymes: elongase, Δ6-desaturase, and Δ5-desaturase [30,31]. Thus, the amount of essential LA and LNA present in the experimental diets may be enough to supply the nutritional requirements of the pirarucu and not negatively affect its growth performance.

Fish require similar nutrients for erythropoiesis as other vertebrates, and they can be affected when fed nutrient-imbalanced diets [32]. Hematocrit, glucose, total protein, cholesterol, and triglycerides are within the range of what has been previously reported for juvenile pirarucu (500 g) fed with 40% crude protein and 8% lipids [33]. In this study, despite the high carbohydrate content (>34%) in the experimental feeds, hyperglycemia was not observed in the juvenile pirarucu. This finding is consistent with the glucose levels reported for pirarucu, which ranged from 45 to 62 mg dL^−1^ [33].

Factors such as age, season, environment, and nutrition are known to influence hemoglobin and serum total protein concentrations in fish [34]. However, in this study, despite the experimental treatments, all these factors were the same during the experiment. In fact, the higher content of DHA from 5 SZ may have influenced the higher hemoglobin levels in the juvenile pirarucu. DHA is a key component of phospholipid biomembranes in fish, which acts by maintaining the fluidity and deformability of the erythrocyte [35]. DHA also influences hemoglobin concentrations, which is crucial for preserving cell shape and function [36]. Although the diet influenced these parameters, both experimental groups remained within the standard hemoglobin range reported for pirarucu [37].

The fatty acid composition of the fillet of the juvenile pirarucu fed the 5 SZ diet reflected that of *Schizochytrium* sp. meal, with elevated concentrations of palmitic acid, DHA, and EPA, and a reduced content of LNA and LA. This pattern has also been observed in freshwater fish fed different dietary lipid sources [17,38,39]. The inclusion of lipids from vegetable sources in aquafeeds has been associated with an increased dietary n-6 fatty acid content, leading to a corresponding reduction in n-3 fatty acids, including EPA and DHA, in fish fillets [40,41]. Excessive n-6 fatty acids are associated with increased inflammation, impaired cardiovascular health [42], and a risk of mental disorders in humans, such as depression and anxiety [43,44]. However, in this study, the n-6 fatty acid content decreased while the n-3 fatty acid content increased in the pirarucu fillets, demonstrating an enhancement in the nutraceutical value of this fish as a food source for humans.

The DHA content in the fillets of the fish fed the 5 SZ diet was 9.45 times higher than in those fed the 0 SZ diet. This result is in accordance with other studies that evaluated the incorporation of DHA in fish fillets through dietary supplementation with *Schizochytrium* sp. meal or oil [17,45,46]. An increase of 5.46 and 2.89 times higher in DHA content was observed in the fillets of channel catfish (*Ictalurus punctatus*) [47] and tambaqui [17], respectively, fed diets containing *Schizochytrium* sp. The high DHA level resulted in a 4.57-fold increase in the n-3/n-6 ratio in the fillets of the pirarucu fed 5 SZ compared to the fish fed 0 SZ. The higher EPA and DHA content, along with the improved n-3/n-6 ratio, in the fillets of the fish fed 5 SZ can provide potential health benefits for humans, as fish is considered one of the most important dietary sources of n-3 fatty acids. These benefits include reducing the inflammatory process and preventing cardiovascular disease, type 2 diabetes, obesity, metabolic syndrome, nonalcoholic fatty liver [42], and mental disorders [43,44].

The World Health Organization (WHO, 2015) [48] recommends a daily intake of 200 mg of EPA + DHA for adults. Considering that wild pirarucu fillets contain approximately 139.5 mg of EPA and DHA per 100 g [21] and farmed pirarucu fillets contain 97.5 mg of EPA and DHA per 100 g [49], it is suggested that 143.37 g of wild pirarucu or 205.13 g of farmed pirarucu fillets are consumed to meet the WHO recommendation. However, with the dietary intervention of *Schizochytrium* sp., pirarucu fillets containing 141.84 mg of EPA + DHA per 100 g of fillet means that just 141.03 g of fillet would be sufficient to meet the same recommendation. The use of *Schizochytrium* sp. as an ingredient in pirarucu diets shows potential for improving the n-3 fatty acid content in fillets, making it a valuable ingredient for aquafeed formulations.

## 5. Conclusions

The inclusion of *Schizochytrium* sp. in diets for pirarucu is an effective strategy to increase the DHA content and improve the n-3/n-6 ratio in the fillet. No adverse effects on pirarucu production performance or health were observed when *Schizochytrium* sp. was included in their diets. Pirarucu fillets enriched with DHA could be marketed as a functional food with a higher nutraceutical value, possibly increasing their market prices.

## Figures and Tables

**Table 1 animals-15-00712-t001:** Feed formulation, proximate, and fatty acid composition of experimental diets fed to pirarucu (*Arapaima gigas*) juveniles for 12 weeks.

Ingredients (%)	*Schizochytrium* sp. Meal ^a^	Experimental Diets ^b^	
		0 SZ	5 SZ
*Schizochytrium* sp. meal ^a^		0	5
Fish meal		21	21
Wheat middling		14.5	14.5
Corn meal		15	13.5
Poultry and offal meal		15	15
Soybean meal		15	15
Meat and bone meal		9.7	9.7
Blood meal		3	0.5
DL-methionine		0.5	0.5
Premix ^c^		3	3
L-lysine		0.18	0.18
Taurine		0.5	0.5
Salt		0.4	0.4
Soybean oil		1	0
Dicalcium phosphate		1	1
BHT		0.20	0.20
Vitamin C		0.02	0.02
*Proximate composition*			
Crude Protein (%)	47.8	40.1	40.2
Crude fat (%)	31.1	9.41	9.32
Carbohydrates ^d^ (%)	18.1	34.5	34.4
Ash (%)	3.02	16.0	16.0
Gross energy ^e^ (MJ kg^−1^)	26.7	17.6	17.7
Fatty acids (% of total lipids)			
*Saturated fatty acids* (SFAs) ^f^			
14:00	3.93	2.06	2.84
16:00	51.45	28.27	37.96
18:00	1.65	11.12	9.3
SFA ^f^	57.03	41.45	50.1
*Monounsaturated*			
16:1 n-7		3.7	3.03
18:1 n-7	0.08	3.4	2.83
18:1 n-9	0.04	27.28	19.72
MUFA ^g^	0.12	34.38	25.58
*Polyunsaturated*			
18:3 n-3	0.05	1.71	0.91
22:4 n-3		0.03	0.07
20:5 n-3	7.78	0.34	1.68
22:6 n-3	34.28	1.56	9.53
n-3 ^h^	42.11	3.64	12.19
18:2 n-6	0.04	18.98	10.68
18:3 n-6	0.03	0.04	0.06
20:4 n-6		0.09	0.08
n-6 ^i^	0.07	19.11	10.82
PUFA ^j^	42.18	22.75	23.01
HUFA ^k^	42.06	2.02	11.36
n-3: n-6	601.57	0.19	1.2

^a^ *Schizochytrium* sp. meal manufactured by ALL-G RichTM products/ALLTECH^®^. ^b^ means of triplicate analyses per sample are shown. ^c^ vitamin and mineral premix per kilogram, including vitamin A (1,000,000 IU), vitamin C (40,000 IU), vitamin D3 (418,350 IU), vitamin E (10,116 IU), vitamin K3 (50 mg), vitamin B1 (1250 mg), vitamin B2 (2500 mg), vitamin B3 (5000 mg), vitamin B5 (250 mg), vitamin B6 (1900 mg), vitamin B7 (80 mg), vitamin B9 (40,000 mg), vitamin B12 (3000 µg), copper (1100 mg), iron (4800 mg), manganese (2100 mg), cobalt (16 mg), iodine (60 mg), zinc (14,000 mg), selenium (60 mg), and inositol (2500 mg). ^d^ carbohydrates (%) = 100 − (protein + lipids + ash). ^e^ gross energy basis on the values calculated for protein, 5.64 kcal/g; lipid, 9.44 kcal/g; and carbohydrate, 4.11 kcal/g (NRC, 2011 [23]). ^f^ total saturated fatty acids also included 12:0, 15:0, 17:0, 20:0, 21:0, 22:0, and 24:0. ^g^ total monounsaturated fatty acids also included 14:1n-7, 15:1n-5, 16:1n-9, 16:1n-7, 17:1n-9 and 21:1n-9. ^h^ total n-3 fatty acids. ^i^ total n-6 fatty acids. ^j^ total polyunsaturated fatty acids also included 22:2n-6. ^k^ total highly unsaturated fatty acids (HUFAs) include all the fatty acids with 20 carbon atoms or more double bonds.

**Table 2 animals-15-00712-t002:** Growth performance and somatic indices of juvenile pirarucu fed with 5% *Schizochytrium* sp. meal.

Variable	Experimental Diets		Statistic	
	0 SZ	5 SZ	PSE	*p*-Value
Survival (%)	98	100	1.41	0.34
Feed intake (g)	13,987	14,376	415.09	0.52
Weigh gain (g)	9007	8941	385.55	0.90
Relative growth ratio (%/day)	1.08	1.07	0.03	0.79
Feed conversion ratio	1.56	1.61	0.091	0.68
Protein efficiency ratio (%)	1.61	1.55	0.091	0.57
Viscerosomatic index (%)	6.64	7.82	0.44	0.09
Hepatosomatic index (%)	1.54	1.74	0.12	0.49
Intraperitoneal fat index (%)	0.43	0.42	0.04	0.89

Abbreviations: 0 SZ: diet with no *Schizochytrium* sp. meal; 5 SZ: diet with 5% *Schizochytrium* sp. meal; PSE: pooled standard error. *p*-values for the nonparametric Kruskal–Wallis test followed by Dunn’s test with 0% for joint ranks.

**Table 3 animals-15-00712-t003:** Hemato-biochemical assays of the pirarucu (*Arapaima gigas*) fed the experimental diets for 12 weeks.

Variables	Experimental Groups		Statistic	
	0 SZ	5 SZ	PSE	*p*-Value
Hemoglobin (g dL^−1^)	7.47 ^b^	8.33 ^a^	0.21	0.007
Hematocrit (%)	36.45	37.8	0.59	0.11
Glucose (mg dL^−1^)	49.36	50.05	3.45	0.96
Total protein (g dL^−1^)	4.47 ^a^	4.26 ^b^	0.05	0.007
Albumin (g dL^−1^)	1.45	1.32	0.16	0.57
Globulin (g dL^−1^)	3.02	2.94	0.16	0.73
Cholesterol (mg dL^−1^)	85.71	104.04	8.03	0.09
Cholesterol HDL (mg dL^−1^)	43.71	44.18	7.60	0.38
Triglycerides (mg dL^−1^)	92.47	102.21	0.37	0.39

Abbreviations: PSE: pooled standard error. *p*-values for the nonparametric Kruskal–Wallis test followed by Dunn’s test with 0% for joint ranks. Lowercase letters indicate significant differences between treatments.

**Table 4 animals-15-00712-t004:** Fatty acid composition (% of total lipids) in the fillets of juvenile pirarucu (*Arapaima gigas*) fed with experimental diets for 12 weeks.

Fatty Acid	Experimental Groups		Statistic	
	0 SZ	5 SZ	PSE	*p*-Value
*Saturated*				
14:00	1.80 ^b^	2.28 ^a^	0.03	0.001
16:00	25.0 ^b^	30.0 ^a^	0.19	0.001
18:00	12.8 ^a^	11.6 ^b^	0.35	0.041
SFA ^a^	42.0 ^b^	45.6 ^a^	0.47	0.01
*Monounsaturated*				
16:1n-7	4.17 ^a^	3.65 ^b^	0.07	0.001
18:1n-7	4.30 ^a^	3.69 ^b^	0.15	0.026
18:1n-9	30.4 ^a^	24.1 ^b^	0.43	0.001
MUFA ^b^	39.6 ^a^	32.0 ^b^	0.5	0.001
*Polyunsaturated*				
18:3 n-3	0.92 ^a^	0.62 ^b^	0.06	0.008
22:4 n-3	0.70 ^a^	0.18 ^b^	0.17	0.001
20:5 n-3	1.17 ^b^	1.87 ^a^	0.11	0.003
22:6 n-3	1.01 ^b^	9.55 ^a^	0.17	0.001
n-3	3.80 ^b^	12.2 ^a^	0.23	0.001
18:2 n-6	13.6 ^a^	9.61 ^b^	0.19	0.001
18:3 n-6	0.69 ^a^	0.27 ^b^	0.02	0.001
20:4 n-6	0.18 ^b^	0.32 ^a^	0.01	0.001
n-6	14.5 ^a^	10.2 ^b^	0.19	0.001
PUFA ^c^	18.3 ^b^	22.4 ^a^	0.3	0.001
HUFA ^d^	3.05 ^b^	11.9 ^a^	0.26	0.001
n-3/n-6	0.26 ^b^	1.19 ^a^	0.21	0.001
Fillet lipids ^e^ (%)	1.67	1.55	-	-

Total unsaturated fatty acids also included 22:0, 21:0, 22:0, and 24:0. Total monounsaturated fatty acids also included 16:1 n-9. ^a^ SFA: sum of all the fatty acids with no double bonds. PSE: pooled standard error. ^b^ MUFA: sum of all the fatty acids with one double bond. ^c^ PUFA: sum of all the fatty acids with no double bonds. ^d^ HUFA: sum of all the fatty acids with 20 carbon atoms or more and 2 or more double bonds. ^e^ Wet basis. Student’s *t*-test *p*-values (*p* < 0.05) indicate a significant difference between the groups. Lowercase letters indicate significant differences between treatments.

## Data Availability

Data from the study are available from the corresponding authors upon reasonable request.

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
