# Peer review of "Dietary Schizochytrium sp. Meal Enhances the Fatty Acid Profile in Pirarucu (Arapaima gigas) Fillets with No Effect on Growth Performance and Health Status"

_animals, 2025, doi:10.3390/ani15050712_

Round 1

Reviewer 1 Report

Comments and Suggestions for Authors

animals-3458963: Dietary Schizochytrium sp. meal enhances the fatty acid profile in pirarucu (Arapaima gigas) fillets with no effect on growth performance and health status

 General comments:

This manuscript presents the effects of Schizochytrium sp. meal inclusion in the diet of juvenile pirarucu and its effects on growth performance, health status, and fillet fatty acid profile. For this purpose, the authors formulated two diets, such as a control and one test diet (with 5% Schizochytrium sp. meal). Authors concluded that incorporating 5% Schizochytrium sp. meal in the diet of juvenile pirarucu: (i) could be an effective strategy to improve DHA content and n3:n-6 ratio in the fillet; (ii) had no adverse effects on fish growth or health; (iii) fish fillets enriched with DHA could be marketed as a functional food with a high nutraceutical value. Overall, it is a good research study and is worthy of investigation. However, I wondered why the authors used only one level of Schizochytrium sp. meal (5%?). I would suggest addressing this in the introduction and discussion sections by providing more information supported by references. After minor revision, this study could contribute important insights to the literature.

Author Response

REVIEWER #1

This manuscript presents the effects of Schizochytrium sp. meal inclusion in the diet of juvenile pirarucu and its effects on growth performance, health status, and fillet fatty acid profile. For this purpose, the authors formulated two diets, such as a control and one test diet (with 5% Schizochytrium sp. meal). Authors concluded that incorporating 5% Schizochytrium sp. meal in the diet of juvenile pirarucu: (i) could be an effective strategy to improve DHA content and n3:n-6 ratio in the fillet; (ii) had no adverse effects on fish growth or health; (iii) fish fillets enriched with DHA could be marketed as a functional food with a high nutraceutical value. Overall, it is a good research study and is worthy of investigation. However, I wondered why the authors used only one level of Schizochytrium sp. meal (5%?). I would suggest addressing this in the introduction and discussion sections by providing more information supported by references. After minor revision, this study could contribute important insights to the literature.

We appreciate your positive feedback on our study and your insightful question regarding the choice of a single inclusion level of Schizochytrium sp. meal. The decision to test only one level (5%) was primarily influenced by the large size of pirarucu (Arapaima gigas), which requires spacious experimental units, and the cost associated with a feeding trial with fish in an advanced life stage. The need for such large units significantly limits the number of dietary treatments that can be feasibly tested within the constraints of available infrastructure and experimental replication. Given these logistical considerations, we opted to evaluate a level that, based on previous studies with other fish species, was expected to yield measurable effects on fillet fatty acid composition and fish performance.

Furthermore, our findings indicate that a 5% inclusion level resulted in significant changes in DHA content and the n-3:n-6 ratio without negatively impacting growth or health parameters. This positive response opens the possibility of further investigations into lower inclusion levels, which could optimize cost-effectiveness while maintaining nutritional benefits.

Reviewer 2 Report

Comments and Suggestions for Authors

This manuscript extends the knowledge regarding biological effects of Schizochytrium sp. as a nutraceutical in fish diets to another aquaculture species, pirarucu (Arapaimas gigas). The impacts of a diet containing 5% Schizochytrium were compared against a control, and the fish measured after 12 weeks for growth, hematology, and fillet quality metrics. The experimental design is appropriate and the results presented are clear. Few differences were observed, with the most important result being a substantial increase in HUFA. I have a few comments for consideration by the authors for improvement of this manuscript, which are itemized below.

The primary argument that the enhanced DHA content may be useful in offsetting impacts of plant protein in aquaculture diets would be strengthened by including more background information on the response of A. gigas to diets formulated with alternative proteins. Lines 79-85 in the introduction imply that there is no problem currently in DHA ratios in farmed pirarucu compared to wild fish, so the potential application of the study is not apparent.

It is not clear what hypothesis underlies the calculation of the corpuscular metrics (MVC, MCH, MCHC), especially given that there was no change in any of these parameters. I do find it interesting that Hb is significantly different, but neither MCH nor MCHC are, but this is not discussed. The authors should either remove these data from the manuscript or improve the text to put their relevance in better context.

There are some consistency problems with the equations given and the data presented in Table 2. The equation for PCE is given, but reported as PER. Similarly, the terminology of viscerosomatic fat index versus intraperiotenal fat should be made consistent between the text and the table (lines 143-156).

Lines 267-272 need to be made more concise as it currently is quite redundant.

Author Response

REVIEWER #2

This manuscript extends the knowledge regarding biological effects of Schizochytrium sp. as a nutraceutical in fish diets to another aquaculture species, pirarucu (Arapaimas gigas). The impacts of a diet containing 5% Schizochytrium were compared against a control, and the fish measured after 12 weeks for growth, hematology, and fillet quality metrics. The experimental design is appropriate and the results presented are clear. Few differences were observed, with the most important result being a substantial increase in HUFA. I have a few comments for consideration by the authors for improvement of this manuscript, which are itemized below.

We appreciate your positive feedback on our study.

The primary argument that the enhanced DHA content may be useful in offsetting impacts of plant protein in aquaculture diets would be strengthened by including more background information on the response of A. gigas to diets formulated with alternative proteins. Lines 79-85 in the introduction imply that there is no problem currently in DHA ratios in farmed pirarucu compared to wild fish, so the potential application of the study is not apparent.

Thank you for your observation. We have added a text to the introduction section to provide sufficient background to clarify the potential application of our study L83-89.

It is not clear what hypothesis underlies the calculation of the corpuscular metrics (MVC, MCH, MCHC), especially given that there was no change in any of these parameters. I do find it interesting that Hb is significantly different, but neither MCH nor MCHC are, but this is not discussed. The authors should either remove these data from the manuscript or improve the text to put their relevance in better context.

Thank you for your observation. We made modifications as suggested.

There are some consistency problems with the equations given and the data presented in Table 2. The equation for PCE is given, but reported as PER. Similarly, the terminology of viscerosomatic fat index versus intraperiotenal fat should be made consistent between the text and the table (lines 143-156).

Thank you for your suggestion. We corrected the equation for PER (Protein efficiency ratio) and terminology of Intraperitoneal fat index (IFI).

Lines 267-272 need to be made more concise as it currently is quite redundant.

Thank you for your suggestion. We corrected as suggested.

Reviewer 3 Report

Comments and Suggestions for Authors

This manuscript (animals-3458963) under the title "Dietary Schizochytrium sp. meal enhances the fatty acid profile in pirarucu (Arapaima gigas) fillets with no effect on growth performance and health status" examined the influence of Schizochytrium sp. meal supplementation as a substitute for soybean oil on growth, nutritional composition, and blood biochemistry parameters in pirarucu. Diets containing Schizochytrium sp. meal had no negative performance in growth and health parameters in this study. Moreover, the diet containing 5% Schizochytrium sp. meal exhibited superior nutritional quality in the pirarucu fillets, resulting in the higher DHA content and n-3:n-6 ratio.

The main content of this paper is important to provide new insights into Schizochytrium sp. meal as the alternatives to fish oil and plant oil in aquaculture. Results from this study also could provide the reference for developing the safe, effective and sustainable terrestrial plant-derived feed ingredients with nutrition-improving properties in pirarucu and other farmed fish.

However, several parts of this manuscript (particularly the introduction and discussion part) are written in an over-long way, showing some ambiguities and awkward syntactic structure/phrasing. Also, there are some mistakes in language, syntax, and format.

Thus, the whole manuscript should be proofread by a professional or native speaker.

Major comments:

1. Regarding the current "Keywords" (Line 40), it did not well match the main content of the paper. Please revise it by adding the correct terms and removing the unnecessary/superfluous phrases. For example, "Schizochytrium sp. meal" and "Arapaima gigas" should be included.

2. The current "1.Introduction" part is not well-organized and has many grammatical errors and statement issues, such as lengthy and confusing sentences.

For example, the original texts of Line 55-62 and Line 62-65 contain the descriptions that seem to be irrelevant. The statements of Line 55-62 (Eicosapentaenoic acid~~of fish fillets.) are closely linked to the main content of 1st paragraph, while Line 62-65 is more associated with the text of next paragraph (Line 66-74).

There is a lack of citations regarding the statements on pirarucu (Line 80-82). These descriptions should be supported by adding relevant references.

Moreover, some sentences that have over-long descriptions should be combined or reworded by shorter sentences. For example, Line 50-54.

Thus, the authors should rephrase the relevant contents of "1.Introduction" for more clearly specifying background in this study.

3. Some critical information in the "2.Materials and Methods" section is insufficient and unclear. Prior to the feeding experiments, experimental fish are cultured for two weeks to adapt to the laboratory conditions. So, what about the main quality parameters of aquatic environment during the acclimating period? Are these parameters were same as those of the feeding trial?

Moreover, it is recommended that the authors provide the sampling process in a separate part.

4. In the "3.Results" part, there are some unnecessary and redundant descriptions. For example, the original texts in Line 234-235 contained unnecessary statements on the results. They could be compressed or deleted directly without any negative influence on the corresponding paragraph. There are similar problems in the other parts of "Results".

Data representation in Table 2-Table 4 is unusual. Generally, data in the published paper are shown as mean ± SD or mean ± SEM. But the value of each parameter in Table 2 is represented as mean. Why? Same issues are present in Table 3-Table 4.

5. Several relevant contents in the "4.Discussion" part are not well-organized and can be incorporated and rephrased. For example, it is suggested to merge the texts of Line 279-286 and Line 287-295 due to the similar themes of these two short paragraphs. . Similar issues on the repeated descriptions are present in Line 271-274.

Moreover, the current discussion part contains some over-long, unnecessary statements and syntactic structure / phrasing issues. For example, in Line 269-272, Line 280-283, Line 320-324, etc., the highlighted phrases can be reworded to avoid any grammatical errors or ambiguity.

Thus, it is recommended that the authors reorganize the main text of "4.Discussion" part for better clarifying your findings in this study. Also, several cited references should be provided in the text of this part.

6. Please check the reference format carefully according to the instructions for authors. The current reference list is chaotic, with publication year, volume and page numbers missing, along with other inconsistencies like DOI vs. non DOI, capitalized vs. lower-case article titles.

For example, in Reference 10, 13, 17, 19, etc, DOI is missing. Furthermore, "https://doi.org/~~~ " or "DOI: ~~", which type is correct in this manuscript? The information on page number is wrong or incomplete in Reference 22, 26, 34, etc. Multiple similar errors are present in the current reference list.

Moreover, the cited literatures published in 2020-2025 are less than 17 (total literatures: 49). Please make sure about 50% of the references are within 5 years (2020-2025).

Thus, the authors need to re-check and modify the reference list seriously.

Other errors (highlighted in yellow) were marked in the PDF file.

So, this manuscript will be reconsidered after major revision.

Author Response

REVIEWER #3

This manuscript (animals-3458963) under the title "Dietary Schizochytrium sp. meal enhances the fatty acid profile in pirarucu (Arapaima gigas) fillets with no effect on growth performance and health status" examined the influence of Schizochytrium sp. meal supplementation as a substitute for soybean oil on growth, nutritional composition, and blood biochemistry parameters in pirarucu. Diets containing Schizochytrium sp. meal had no negative performance in growth and health parameters in this study. Moreover, the diet containing 5% Schizochytrium sp. meal exhibited superior nutritional quality in the pirarucu fillets, resulting in the higher DHA content and n-3:n-6 ratio.

The main content of this paper is important to provide new insights into Schizochytrium sp. meal as the alternatives to fish oil and plant oil in aquaculture. Results from this study also could provide the reference for developing the safe, effective and sustainable terrestrial plant-derived feed ingredients with nutrition-improving properties in pirarucu and other farmed fish.

However, several parts of this manuscript (particularly the introduction and discussion part) are written in an over-long way, showing some ambiguities and awkward syntactic structure/phrasing. Also, there are some mistakes in language, syntax, and format.

Thus, the whole manuscript should be proofread by a professional or native speaker.

Thank you for your feedback. We have carefully revised the introduction and discussion sections to remove redundancies. However, we also made an effort to incorporate all suggested changes, including those from other reviewers who requested additional information. Regarding the comments on language, syntax, and format, although we are native Portuguese speakers, our manuscript was reviewed before the first submission by Dr. Yamamoto, a co-author and Brazilian-born U.S. faculty member, as well as by a professional native English speaker.

Major comments:

  1. Regarding the current "Keywords" (Line 40), it did not well match the main content of the paper. Please revise it by adding the correct terms and removing the unnecessary/superfluous phrases. For example, "Schizochytrium sp. meal" and "Arapaima gigas" should be included.

Thank you for your recommendation. We corrected as suggested.

  1. The current "1.Introduction" part is not well-organized and has many grammatical errors and statement issues, such as lengthy and confusing sentences.

For example, the original texts of Line 55-62 and Line 62-65 contain the descriptions that seem to be irrelevant. The statements of Line 55-62 (Eicosapentaenoic acid~~of fish fillets.) are closely linked to the main content of 1st paragraph, while Line 62-65 is more associated with the text of next paragraph (Line 66-74).

Thank you for your recommendation. We corrected as suggested.

There is a lack of citations regarding the statements on pirarucu (Line 80-82). These descriptions should be supported by adding relevant references.

Thank you for your observation. We made modifications as suggested.

Moreover, some sentences that have over-long descriptions should be combined or reworded by shorter sentences. For example, Line 50-54.

Thank you for your recommendation. We reworded as suggested

Thus, the authors should rephrase the relevant contents of "1.Introduction" for more clearly specifying background in this study.

Thank you for your recommendation. We rephrased as suggested

  1. Some critical information in the "2.Materials and Methods" section is insufficient and unclear. Prior to the feeding experiments, experimental fish are cultured for two weeks to adapt to the laboratory conditions. So, what about the main quality parameters of aquatic environment during the acclimating period? Are these parameters were same as those of the feeding trial?

Moreover, it is recommended that the authors provide the sampling process in a separate part.

Thank you for your recommendation. The experiment was conducted in a recirculating system with aquatic plants for biological filtration. The water quality parameters were stable throughout the acclimation period and the feeding trial.

  1. In the "3.Results" part, there are some unnecessary and redundant descriptions. For example, the original texts in Line 234-235 contained unnecessary statements on the results. They could be compressed or deleted directly without any negative influence on the corresponding paragraph. There are similar problems in the other parts of "Results".

Thank you for your recommendation. We corrected as suggested.

Data representation in Table 2-Table 4 is unusual. Generally, data in the published paper are shown as mean ± SD or mean ± SEM. But the value of each parameter in Table 2 is represented as mean. Why? Same issues are present in Table 3-Table 4.

Thank you for your insightful comments. The Pooled Standard Error (PSE) was chosen to represent the variability within the treatments because it integrates the variability of both experimental groups (5SZ and 0SZ), providing a more robust estimate of the error compared to traditional methods such as standard deviation (SD) or standard error of the mean (SEM), which reflect the dispersion within each group separately. By using the PSE, we are able to calculate a single value, which gives us a more accurate view of the difference between the treatments, taking into account the combined variability of the groups, which is particularly useful for comparing the means of treatments with similar characteristics.

  1. Several relevant contents in the "4.Discussion" part are not well-organized and can be incorporated and rephrased. For example, it is suggested to merge the texts of Line 279-286 and Line 287-295 due to the similar themes of these two short paragraphs. . Similar issues on the repeated descriptions are present in Line 271-274.

Thank you for your recommendation. We reworded as suggested.

Moreover, the current discussion part contains some over-long, unnecessary statements and syntactic structure / phrasing issues. For example, in Line 269-272, Line 280-283, Line 320-324, etc., the highlighted phrases can be reworded to avoid any grammatical errors or ambiguity.

Thank you for your recommendation. We reworded as suggested.

Thus, it is recommended that the authors reorganize the main text of "4.Discussion" part for better clarifying your findings in this study. Also, several cited references should be provided in the text of this part.

Thank you for your recommendation. We reorganized as suggested.

  1. Please check the reference format carefully according to the instructions for authors. The current reference list is chaotic, with publication year, volume and page numbers missing, along with other inconsistencies like DOI vs. non DOI, capitalized vs. lower-case article titles.

For example, in Reference 10, 13, 17, 19, etc, DOI is missing. Furthermore, "https://doi.org/~~~ " or "DOI: ~~", which type is correct in this manuscript? The information on page number is wrong or incomplete in Reference 22, 26, 34, etc. Multiple similar errors are present in the current reference list.

Moreover, the cited literatures published in 2020-2025 are less than 17 (total literatures: 49). Please make sure about 50% of the references are within 5 years (2020-2025).

Thus, the authors need to re-check and modify the reference list seriously.

Thank you for your observation. We made modifications as suggested.

Other errors (highlighted in yellow) were marked in the PDF file.

Thanks for your observation. We corrected as suggested.

So, this manuscript will be reconsidered after major revision.

Reviewer 4 Report

Comments and Suggestions for Authors

The manuscript titled "Dietary Schizochytrium sp. meal enhances the fatty acid profile in pirarucu (Arapaima gigas) fillets with no effect on growth performance and health status" provides applied insights relevant to the aquaculture sector. However, several areas require clarification and improvement. Below are my suggestions and observations to be addressed: 

The manuscript should include a comprehensive overview of Schizochytrium sp., including its key attributes, such as nutritional composition (e.g., crude protein, DHA, and other essential nutrients), its relevance and potential as a sustainable aquaculture feed ingredient and why it was chosen for this study. 

The manuscript mentions that Schizochytrium sp. contains 47.8% crude protein. However, the protein levels in the 0SZ and 5SZ diets are reported to be similar. This raises concerns about potential errors in formulation or estimation. This suggests potential errors in diet formulation or estimation, which need to be addressed. The authors must provide a clear explanation of how protein content remained unchanged despite the different inclusion of Schizochytrium sp in the diet.

Multiple ingredients were altered in the formulation, how did the authors conclude the effect of 0SZ vs. 5SZ, which appears to be the primary variable of interest? This design flaw impacts the validity of the interpretations made.

Provide complete details about the production conditions, growth requirements, and harvesting process for Schizochytrium sp. used in this study.

Include information about the control diet composition, as it is essential for understanding the experimental framework.

Justify the selection of only two levels of Schizochytrium sp. inclusion (0SZ and 5SZ). Were intermediate or higher inclusion levels considered? If not, why?

The current data analysis approach seems insufficient for supporting the conclusions drawn. The statistical analysis needs revision. A one-way ANOVA should be employed for greater clarity, and p-values should be reported in all tables to enhance the robustness of the findings.

Upon reviewing the fatty acid profiles of the feed and the fillet, there appears to be no significant retention of DHA in the fillets. This contradicts the claim that DHA content increased in the fish fillet.

The authors need to redo their interpretation of the results and provide clear evidence to support their conclusions regarding DHA enrichment.

The conclusion appears to be based on flawed data analysis and misinterpretation of the results. A thorough review of the methodology, statistical analysis, and findings is necessary to ensure the validity of the study claims.

Author Response

REVIEWER #4

The manuscript titled "Dietary Schizochytrium sp. meal enhances the fatty acid profile in pirarucu (Arapaima gigas) fillets with no effect on growth performance and health status" provides applied insights relevant to the aquaculture sector. However, several areas require clarification and improvement. Below are my suggestions and observations to be addressed:

The manuscript should include a comprehensive overview of Schizochytrium sp., including its key attributes, such as nutritional composition (e.g., crude protein, DHA, and other essential nutrients), its relevance and potential as a sustainable aquaculture feed ingredient and why it was chosen for this study.

Thanks for your observation. We corrected it as suggested.

The manuscript mentions that Schizochytrium sp. contains 47.8% crude protein. However, the protein levels in the 0SZ and 5SZ diets are reported to be similar. This raises concerns about potential errors in formulation or estimation. This suggests potential errors in diet formulation or estimation, which need to be addressed. The authors must provide a clear explanation of how protein content remained unchanged despite the different inclusion of Schizochytrium sp in the diet.

Thanks for your observation. We apologize for the mistake committed in the concentration of meat and bone meal for the control diet. The right concentration is similar to the 5% of Schizochytrium sp. meal inclusion (9.7%), it was a typing error in the data. In addition, During the diets formulation, the substitution of soybean oil for Schizochytrium sp. meal imbalanced the crude protein levels in both diets. Because of this, in order to balance the diets, some protein ingredients were lightly changed, except blood meal, which had a significant influence because of its high protein levels (88.21%).

Multiple ingredients were altered in the formulation, how did the authors conclude the effect of 0SZ vs. 5SZ, which appears to be the primary variable of interest? This design flaw impacts the validity of the interpretations made.

Thank you for your observation. Both diets were formulated to be isonitrogenous (40% crude protein), isolipidic (9%), and isoenergetic (17.65 MJ kg-1), ensuring that the primary variables affecting the results were the inclusion levels of Schizochytrium sp. (SZ) meal (0SZ vs. 5SZ). Although other ingredients were adjusted to maintain the overall nutritional balance, we focused on isolating the effect of the SZ meal inclusion. This was achieved by ensuring that the nutritional profile remained consistent between the two diets and meeting the nutritional requirements established for this species. The key distinction being the treatments was the inclusion of SZ meal at different levels.

The DHA content and fatty acid profile in pirarucu are directly attributable to the inclusion of SZ meal, as these were the primary factors altered between the two diets. The careful formulation of the diets allowed us to confidently conclude that the differences in the outcome measures were due to the variable of interest, SZ meal inclusion, and not the other ingredients.

Provide complete details about the production conditions, growth requirements, and harvesting process for Schizochytrium sp. used in this study.

The Schizochytrium sp. meal was kindly donated by Alltech Inc. Production conditions, growth requirements and harvesting process were not informed to the authors. Nonetheless, the ingredient was analyzed for proximate composition prior to formulate the diets.

Include information about the control diet composition, as it is essential for understanding the experimental framework.

Thank you for your observation. We made modifications as suggested.

Justify the selection of only two levels of Schizochytrium sp. inclusion (0SZ and 5SZ). Were intermediate or higher inclusion levels considered? If not, why?

We appreciate your positive feedback on our study and your insightful question regarding the choice of a single inclusion level of Schizochytrium sp. meal. The decision to test only one level (5%) was primarily influenced by the large size of pirarucu (Arapaima gigas), which requires spacious experimental units and it has a large associated cost for diet manufacturing and maintaining the animals. The need for such large units significantly limits the number of dietary treatments that can be feasibly tested within the constraints of available infrastructure and experimental replication. Given these logistical considerations, we opted to evaluate a level that, based on previous studies with other fish species, was expected to yield measurable effects on fillet fatty acid composition and fish performance.

Furthermore, our findings indicate that a 5% inclusion level resulted in significant changes in DHA content and the n-3:n-6 ratio without negatively impacting growth or health parameters. This positive response opens the possibility of further investigations into lower inclusion levels, which could optimize cost-effectiveness while maintaining nutritional benefits.

The current data analysis approach seems insufficient for supporting the conclusions drawn. The statistical analysis needs revision. A one-way ANOVA should be employed for greater clarity, and p-values should be reported in all tables to enhance the robustness of the findings.

Thank you for your suggestion. However, I would like to respectfully disagree with the recommendation. A one-way ANOVA is a statistical test used to assess whether there are significant differences among the means of three or more independent groups. However, in experiments involving only two groups, a one-way ANOVA can still be used without loss of generality, as the F value obtained in this case is equivalent to the t value from the Student's t-test. The F-test calculates the ratio between the variability between groups and the variability within groups, and when applied to two groups, it results in the same information provided by the t-test. The Student's t-test, however, is particularly appropriate for experiments with two groups, as in our case (0SZ vs. 5SZ), and is widely accepted in such experimental designs. It is a robust statistical test for comparing the means of two independent groups. Given that our study compares only two dietary treatments, the t-test provides a precise and efficient means to analyze the differences between them. Therefore, we believe that using the Student's t-test is the most suitable approach for this study.

Additionally, we ensured that the assumptions of the t-test were met by conducting normality (Shapiro-Wilk) and homogeneity of variances (Levene) tests before performing the analysis.

Regarding the presentation of results, we have revised the tables to ensure that all p-values are explicitly reported, enhancing the clarity and transparency of our findings.

Upon reviewing the fatty acid profiles of the feed and the fillet, there appears to be no significant retention of DHA in the fillets. This contradicts the claim that DHA content increased in the fish fillet.

Thanks for your observation. We respectfully disagree with your assessment regarding DHA retention in the fish fillets. As presented in our results, the fillets of fish fed the diet containing 5% Schizochytrium sp. meal presented a DHA content of 9.55% of total lipids, compared to 1.01% of total lipids in the control group. This represents 845% increase in DHA content relative to the control diet. These values clearly demonstrate a substantial increase in DHA accumulation in the fish fillet, supporting the claim made in our manuscript.

The authors need to redo their interpretation of the results and provide clear evidence to support their conclusions regarding DHA enrichment.

Thank you for your recommendation. The Authors double checked the results.

The conclusion appears to be based on flawed data analysis and misinterpretation of the results. A thorough review of the methodology, statistical analysis, and findings is necessary to ensure the validity of the study claims.

Thank you for your recommendation. We have carefully reviewed the methodology, statistical analysis, and results, and we have revised the conclusion accordingly.

Round 2

Reviewer 3 Report

Comments and Suggestions for Authors

The revised manuscript (ID: animals-3458963) entitled "Dietary Schizochytrium sp. meal enhances the fatty acid profile in pirarucu (Arapaima gigas) fillets with no effect on growth performance and health status" has been modified as suggested. The authors reply to the reviewer’s comments one by one (online system).

The current revision is suitable for publication in your journal, though it still contains some minor errors, such as the format of reference list. For example, in Reference 38 there is no DOI information. Similarly, the information on page number is missing in Reference 44. In Reference 1, 3, 9, 11, etc., DOI is presented inconsistently and wrongly, showing "https://doi: ~~". Additionally, "https://doi.org/~~~" or "https://dx.doi.org/~~~" which one is correct?

Thus, it is recommended to accept this paper although there are some minor mistakes.

Reviewer 4 Report

Comments and Suggestions for Authors

The manuscript has been substantially improved, and I can see that the authors have addressed all the queries raised during the review.

Comments on the Quality of English Language

The manuscript has been substantially improved after revision. I recommend manuscript for publication in its current form.